# Is Obesity a Risk Factor for Loss of Reduction in Children with Distal Radius Fractures Treated Conservatively?

**DOI:** 10.3390/children9030425

**Published:** 2022-03-17

**Authors:** Andrea Vescio, Gianluca Testa, Marco Sapienza, Alessia Caldaci, Marco Montemagno, Antonio Andreacchio, Federico Canavese, Vito Pavone

**Affiliations:** 1Department of General Surgery and Medical Surgical Specialties, Section of Orthopaedics and Traumatology, University Hospital Policlinico-Vittorio Emanuele, University of Catania, 95123 Catania, Italy; andreavescio88@gmail.com (A.V.); gianpavel@hotmail.com (G.T.); marcosapienza09@yahoo.it (M.S.); alessia.c.92@hotmail.it (A.C.); m-acor@hotmail.it (M.M.); 2Department of Pediatric Orthopedic Surgery, “V. Buzzi” Children Hospital, 20154 Milan, Italy; prof.andreacchio@gmail.com; 3Pediatric Orthopedic Surgery Service, Hôpital Jeanne de Flandre, University of Lille, 59037 Lille, France; canavese_federico@yahoo.fr

**Keywords:** childhood obesity, distal radius fracture, secondary displacement, loss of reduction, conservative treatment

## Abstract

Background: Obesity in children is a clinical and social burden. The distal radius (DR) is the most common site of fractures in childhood and conservative treatment is widely used. Loss of reduction (LOR) is the major casting complication. The aim of this study is to evaluate obesity as a risk factor for LOR in children with displaced DR fractures (DRF) treated conservatively. Methods: 189 children under 16 years of age were treated conservatively for DRF. Patients were divided into three groups: normal weight (NW), overweight (OW) and obese (OB). The following radiographic criteria were evaluated in all patients: amount of initial translation (IT); quality of initial reduction; Cast (CI), Padding (PI), Canterbury (CaI), Gap (GI) and Three-Points (3PI) indices and the presence of LOR. Results: Statistically significant differences were found between the NW and the OB group for number of LOR (*p* = 0.002), severity (grade) of initial translation (*p* = 0.008), quality of initial reduction (*p* = 0.01) as well as CsI and CaI (*p* < 0.001). Conclusions: Obese children have a significantly higher rate of LOR compared to NW and OW children. A close follow-up is necessary in this population of patients. Preventive percutaneous pinning could be considered in older obese patients in order to reduce the need for further treatment.

## 1. Introduction

Obesity in childhood has become one of the most significant public health complications in numerous nations around the world [1], as it has led to the emergence of several grave obesity-related comorbidities [2,3]. In particular, both overweight and obese children have an altered cortical trabecular bone microarchitecture, which may increase their propensity to fracture following trauma [4]. In addition, recent research has highlighted the role played by leptin in altering the microstructural properties of bone in obese children, which may further increase the propensity to fracture in this subpopulation of patients [5]. Distal radius fractures (DRF) represent the most common type of fracture in children and account for 25% to 43% of all fractures in children [6,7,8]. According to Rana et al., obese patients tend to have a higher prevalence of traumatic limb lesions and a higher rate of operative treatment, despite the fact that cast treatment is the recommended treatment option in most cases [9]. On the other hand, open fractures, irreducible fractures, fractures with unacceptable alignment after attempted closed reduction, displaced intra-articular fractures, fractures with loss of reduction, as well as floating elbow injuries are candidates for surgical management [10].

Loss of reduction (LOR) is the most frequent complication in displaced DRF managed conservatively by closed reduction and casting (CRC) in children and adolescents [11]. LOR ranges from 21% to 47% and can be influenced by ‘fracture-related’ factors (displacement at fracture site, fracture obliquity and residual angulation after closed reduction) [12,13,14,15] and/or ‘surgeon-related’ factors, such as the Cast Index (CsI), the Padding Index (PI), the Canterbury Index (CAI), the Gap Index (GI) or the 3-points Index (3PI) [16,17,18,19,20,21]. In particular, PI, CAI, GI and 3PI have been shown to be optimal predictors of secondary displacement in children with DRF treated conservatively [22,23,24]. Other authors have reported that among ‘fracture-related’ factors, obesity enhances the risk of LOR after closed reduction in both DRF and forearm fractures in children and adolescents [22,23,24]. However, despite the findings of previous studies, the role of childhood obesity has not yet been clearly defined.

The aims of this study were (1) to evaluate obesity as a predictor of DRF pattern severity and of a low quality of fracture reduction in children; (2) to assess childhood obesity as a potential risk factor for secondary displacement in children with DRF treated conservatively by CRC; (3) to evaluate radiological parameters (CsI, PI, CAI, GI and 3PI) in different weight cohorts.

For obese DRF-affected children, a greater severity of fracture patterns, a lower quality of reduction, and greater difficulties in proper casting can be predicted

## 2. Materials and Methods

### 2.1. Sample Selection

Medical records of DRF-affected children under the age of 16 treated with closed reduction and long cast immobilization from January 2012 to December 2017 were reviewed retrospectively. 

The inclusion criteria were as follows: (1) skeletal immaturity; (2) confirmed diagnosis of displaced DRF (>10° angulation on lateral radiographs; >20° angulation on AP radiographs; <50% apposition on AP or lateral radiographs); (3) chronological age under 16 years; (4) treatment at the emergency department as per the ‘treatment protocol’ described below; and (5) complete radiographic data.

Gender, age at the time of injury, mechanism of accident, involved side, height, weight, Body Mass Index (BMI), presence or absence of associated neurovascular injury and whether it was closed or open, as well as length of post-operative immobilization of all patients admitted to the emergency department were collected (Table 1). According to the WHO BMI-for-age criteria (5–19 years) [24], the sample was divided into 3 cohorts: normal weight (NW), overweight (OW) and obese (OB) [25,26] (Table 2).

Specifically, the OB group included patients with a BMI that was 2 or more standard deviations (SD) higher than that in the normal population (equivalent to a BMI of 30 kg/m^2^ at 19 years), while the OW group included patients with a BMI that was 1 SD higher than in the normal population (equivalent to a BMI of 25 kg/m^2^ at 19 years).

The WHO has published references for children and adolescents aged 5 to 19 and has suggested cut-off values for overweight and obesity based on z-scores [26]. WHO BMI-for-age standards are more precise than BMI in children due to their strict dependence on age and gender [25].

### 2.2. Treatment Protocol

The same experienced pediatric orthopedic surgical team treated all patients at the emergency department of our institution. Prior to closed reduction and cast immobilization, patients were administered equimolecular N_2_O_2_. A maximum of two attempts for a proper reduction were admitted before surgical treatment. After the reduction, a long arm cast was applied for 6 weeks, and the elbow was immobilized at 90–100° of flexion with the forearm in a neutral position. 

### 2.3. Radiographic Evaluation

Radiographies of the injured forearm from the anteroposterior (AP) and lateral views, at 1, 2 and 4 weeks after reduction, were used to evaluate the fracture location, the direction and amount of displacement, as well as consolidation.

Radiographic standards for loss of reduction were defined as: <50% apposition on either radiograph, or >15° change on the lateral radiograph; >20° angulation on the lateral radiograph; and >10° angulation on the AP radiograph at the 1-week assessment [15]. According to the criteria of Mani et al. [15,27], the translations were classified as follows: grade I, no translation; grade II, translation <50%; grade III, translation >50%; and grade IV, complete loss of contact between fragments. According to the criteria proposed by Alemdaroglu et al., the quality of the initial reduction was divided in to (a) “anatomic reduction” without any remaining translation or angulation; (b) “good reduction” with less than 10° of residual dorsal angulation or 2 mm of residual translation; and (c) “fair reduction” with 10° to 20° residual angulation or a residual translation of 2–5 mm [15,28]. 

Moreover, the CsI, PI, CaI, GI, and 3PI were calculated for all subjects following conservative treatment.

On the AP radiographs at the level of the fracture site, the proportion between the inner diameter of the cast on lateral radiographs and the inner diameter of the cast was defined as the CsI; the risk of LOR is higher if the ratio is >0.7 [29].

On AP radiographs, the proportion between a) the inner cast to bone/skin distance on lateral radiographs and b) the maximum interosseous distance of the ratio was defined as the PI; the risk of LOR is higher if the ratio is >0.3 [30].

The summation of the CsI and PI results in the CaI; if the CaI is higher than 1.1, the risk of LOR is advanced [30].

The proportion between the inner plaster diameter on AP radiographs and lateral radiographs at the fracture site results in the GI; if the GI is higher than 0.15, the risk of LOR is advanced [31]. The 3PI takes into account the gap at the fracture site and the gap proximal and distal to the fracture; for values over 0.8, the risk of LOR is advanced [15].

### 2.4. Statistical Analysis

Continuous data are reported as means and standard deviations. In order to compare age means between the three cohorts of subjects, ANOVAs were performed. χ^2^-tests were used to evaluate the homogeneity of the three groups based on age, gender and laterality, and to compare the cut-off values of each index. Odds Ratios (OR) were used to calculate the risk factor for LOR. The selected threshold for statistical significance was *p* < 0.05. All statistical analyses were performed using 2016 GraphPad Software (GraphPad Inc., San Diego, CA, USA).

## 3. Results

### 3.1. Sample

In total, 189 (109 males; 80 females) out of 278 patients met the inclusion criteria and were enrolled in the trial. 

The NW group contained 67 patients (35.5%; 38 boys, 29 girls) with a mean age of 11.6 ± 1.4 years at the time of injury (range, 10.2–14.9) and a mean WHO BMI-for-age of 0.8 ± 0.4 (range, 0.1–0.9); the OW group contained 84 patients (44.4%; 46 boys, 38 girls) with a mean age of 12.1 ± 2.3 years (range, 9.9–15.2) and a mean WHO BMI-for-age of 1.6 ± 0.5 (range 1.0–1.9); and the OB group contained 38 patients (20.1%; 25 boys, 13 girls) with a mean age of 11.3 ± 1.6 years (range 10.1–14.9) and a mean WHO BMI-for-age of 2.2 ± 0.2 (range 2.1–2.8). The three groups of patients did not differ significantly in their demographics (*p* > 0.05), except for WHO BMI-for-age (*p* < 0.05) (Table 1).

### 3.2. Radiographic Evaluation

Out of 189 patients, 41 (21.7%) had a LOR; of these, 23/41 (56.1%) required secondary surgical treatment (closed reduction and percutaneous pinning), while the remaining 18 underwent CRC again (43.9%).

The OR of secondary displacement was 0.44 (95% C.I.; 0.19–0.98) in the NW group, 0.86 (95% C.I.; 0.43–1.73) in the OW group, and 3.14 (95% C.I.; 1.44–6.81) in the OB group.

Statistically significant differences were found between the NW and the OB group for the number of LOR (*p* = 0.002), severity (grade) of initial translation (*p* = 0.008), and quality of initial reduction (*p* = 0.01). On the other hand, for NW and OW patients, the number of LOR, the severity of initial translation and the initial reduction quality were comparable (*p* > 0.05). There was a significant difference for the number of LOR between the OW and OB groups (*p* = 0.02) (Table 2). 

According to the radiographic criteria, no statistically significant differences were found (*p* > 0.05) between the three groups of patients, except for the CsI between NW and OB patients (*p* < 0.00001) and between OW and OB patients (*p* = 0.001), and for the CAI between NW and OB patients (*p* = 0.001) and between OW and OB patients (*p* = 0.04) (Table 3).

## 4. Discussion

According to our findings, a worse initial translation and worse quality of closed reduction were recorded in conservatively treated DRF for the group of obese children compared to overweight children and children of normal weight. Moreover, obese children were shown to have a higher risk of LOR compared to overweight subjects and those of normal weight. This report also found that proper cast molding is more difficult in obese children and that this can potentially lead to LOR. Overall, only 12.2% of children required secondary surgical treatment and the majority of them were obese.

Childhood obesity is associated with multiple medical comorbidities, including several orthopedic conditions [32,33]; in addition, a strong correlation has been reported between fat mass and pediatric trauma, particularly for lower limb and forearm fractures [9]. A significant association of obesity with increased rates of open reduction internal fixation (ORIF) for distal radius and ulna fractures has been reported, and obese patients have also been shown to exhibit significant increases in complications after undergoing ORIF [34]. For these reasons, proper casting is mandatory, despite LOR following CRC being a relatively frequent complication of pediatric forearm and wrist fractures [10,15]. In a recent metanalysis, Vescio et al. [35] reported that the risk of LOR was four times higher in obese patients compared to subjects of normal weight. 

Our data showed that 21.7% of patients had LOR, with 78% of cases being OW (17 out of 41; 41.5%) and OB patients (15 out of 41; 36.6%). In particular, the risk of LOR was 3.1 times higher for OB patients compared to the rest of the sample (OW and NW patients combined); 2.6 times higher (95% C.I.; 1.11–5.96) compared to the OW group, and 4.2 times higher (95% C.I.; 1.61–10.95) compared to the NW group. These findings are comparable to those reported by Okoroafor [23], DeFrancesco [24] and Liu et al. [33]. These findings are probably causally related to the large soft tissue envelope in obese children that makes it harder to maintain a stable reduction [15,23] (Figure 1). 

It can be assumed that a lower reduction quality is related to a greater severity of initial translation. 

Auer et al. reported that obese children with distal radius and forearm fractures tended to have a poorer quality of reduction and were at increased risk of LOR following CRC [22]. In our series, 42.1% of obese children had a fair reduction in the fracture. 

We found that obese children had more severe initial translation compared to NW and OW patients (*p* = 0.008), as previously reported by Mani et al. [27]; the findings in children were comparable to those in adult patients [36,37]. This finding may be linked to the correlation between two key aspects, the surplus of fat mass and an amplified force applied to bone tissue at the time of injury.

In particular, an excess of fat mass has been shown to have a potentially detrimental effect on bone development in children [38], while an increased force applied to the skeleton increases the risk of fractures and the severity of displacement [34]. 

Both factors can interfere with proper fracture reduction, as the key components to achieve fracture stabilization with conservative treatment are proper cast molding, thin and uniform padding, and three-point fixation [13] (Figure 2).

In order to assess the effect of childhood obesity on the conservative treatment of DRF in children, we performed a radiological assessment of the major predictive indexes such as the CsI, PI, CAI, GI and 3PI. To the best of our knowledge, this is the first study to consider surgeon-related factors in overweight and obese patients from a radiological perspective and to compare them to children of normal weight. According to a recent study, the CsI, PI, and CAI were the predictive indexes most commonly used to estimate the risk of LOR in children with DRF treated conservatively [15]. 

According to our findings, NW, OW and OB patients showed no statistically significant differences for the PI and 3PI. On the other hand, according to the CsI and CAI, overweight and obese children had a higher incidence of inadequate cast molding compared to children with normal weight (*p* < 0.05). An abnormal proportion of the plaster diameter in the lateral and anteroposterior planes due to the excess of fat mass poorly affecting cast molding, lead to an amplified risk of LOR [13]. Hanstein et al. reported a weak positive correlation for distal radius fractures only (r = 0.171, *p* = 0.037) between the CsI and the BMI-for-age percentile [39].

The limitations of this study are its retrospective nature, the absence of clinical evaluation, the radiological follow-up being limited to the short term, and no long-term evaluations being performed. Moreover, there was possible bias regarding objectively measurable parameters.

## 5. Conclusions

In conclusion, long-arm casting represents a simple, safe, effective, and inexpensive option for the management of displaced DRF in children. However, obese children require proper cast molding even though it is more difficult to achieve compared to overweight and normal weight patients. As a consequence, obese children require close clinical and radiographic follow-ups [22]. Obesity should be considered as a negative predictive factor for the conservative treatment of DRF in children; in older obese children, percutaneous pinning could be considered as a primary treatment option.

## Figures and Tables

**Figure 1 children-09-00425-f001:**
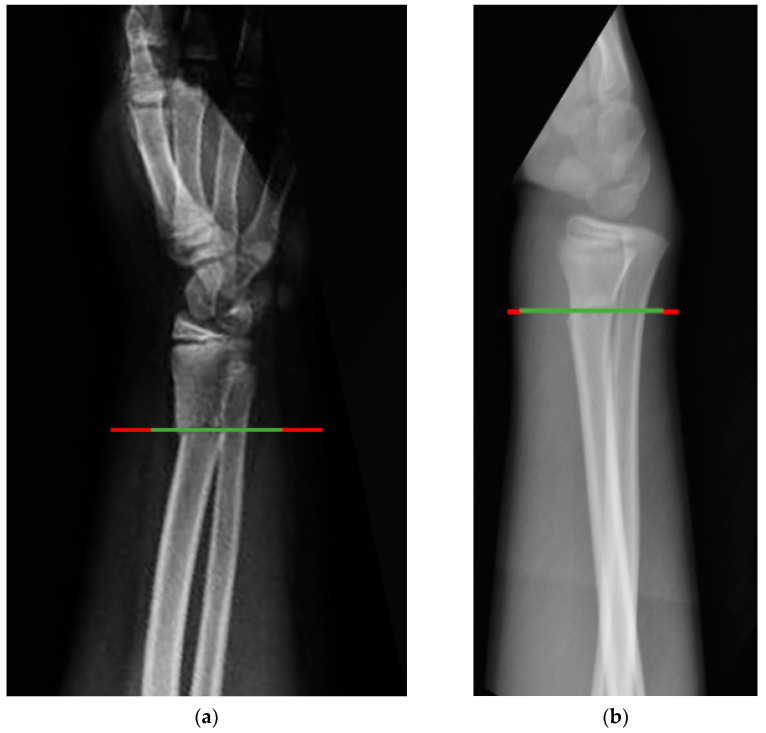
A comparison of the lateral view of the fat layer (red lines) at a fracture site (green lines) between (**a**) an obese patient and (**b**) a patient of normal weight matched for age and gender.

**Figure 2 children-09-00425-f002:**
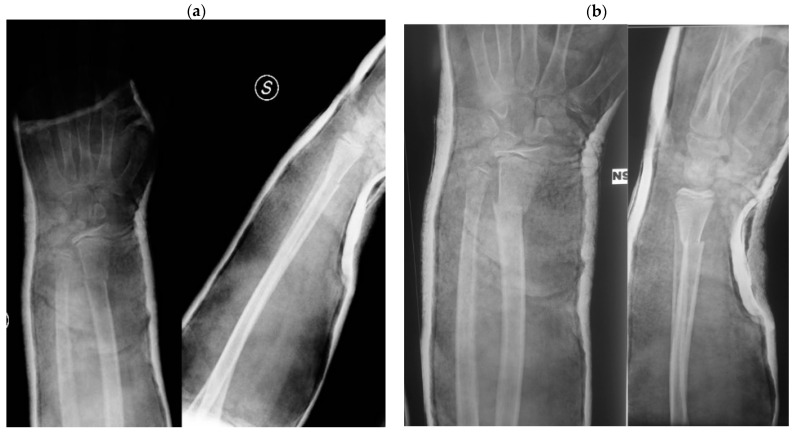
Case report of an obese patient. (**a**) Post-reduction casting X-ray; (**b**) one-week post-reduction X-ray with loss of reduction; (**c**) surgical treatment of the fracture.

**Table 1 children-09-00425-t001:** Patients’ characteristics.

Group	Patients	M	F	WHO BMI-for-Age	Mean Age	Side	
						Right	Left
Sample	189	109	80	1.4 ± 0.4	11.8 ± 1.8	128	61
NW	67	38	29	0.8 ± 0.4	11.6 ± 1.4	45	22
OW	84	46	38	1.6 ± 0.5	12.1 ± 2.3	59	25
OB	38	25	15	2.2 ± 0.2	11.3 ± 1.6	24	14

**Table 2 children-09-00425-t002:** Loss of reduction (LOR), grade of initial translation, and quality of initial reduction for each cohort.

Group	LOR		Grade of Initial Translation				Quality of Initial Reduction		
	No	Yes	1	2	3	4	Anatomic	Good	Fair
Sample	148(78.3)	41(21.7)	44(23.3)	55(29.1)	57(30.1)	33(17.5)	57(78.3)	80(78.3)	52(78.3)
NW	58(78.3)	9(78.3)	23(78.3)	21(78.3)	16(78.3)	7(78.3)	26(78.3)	30(78.3)	11(78.3)
OW	67(78.3)	17(78.3)	16(78.3)	26(78.3)	28(78.3)	14(78.3)	22(78.3)	37(78.3)	25(78.3)
OB	23(78.3)	15(78.3)	5(78.3)	8(78.3)	13(78.3)	12(78.3)	9(78.3)	13(78.3)	16(78.3)

**Table 3 children-09-00425-t003:** Radiological indices for each cohort.

	Cast Index	Padding Index	Canterbury Index	Gap Index	Three-Point Index
	(Cutoff/Tot)	(Cutoff/Tot)	(Cutoff/Tot)	(Cutoff/Tot)	(Cutoff/Tot)
NW Group	45/67 (67.2%)	52/67 (77.6%)	48/67 (71.6%)	38/67 (56.7%)	43/67 (64.2%)
*p* NW vs. OW	0.06	0.48	0.51	0.81	0.87
OW Group	54/84 (64.3%)	61/84 (72.6%)	56/84 (66.7%)	46/84 (54.8%)	55/84 (65.5%)
*p* OW vs. OB	0.001	0.44	0.04	0.87	0.80
OB Group	12/38 (31.6%)	25/38 (65.8%)	18/38 (47.4%)	20/38 (52.6%)	24/38 (63.2%)
*p* NW vs. OB	<0.00001	0.19	0.01	0.69	0.92

## Data Availability

Not applicable.

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
