# Peer review of "Is Obesity a Risk Factor for Loss of Reduction in Children with Distal Radius Fractures Treated Conservatively?"

_children, 2022, doi:10.3390/children9030425_

Round 1

Reviewer 1 Report

Please shorten the title of the article.
Please clearly state the purpose of this research.
Please give a hypothesis.
Please provide the bioethics committee number.
The research results should be expanded.

Author Response

Reviewer 1

Thank for your suggestion and contribution to the manuscript, after the revision of the article we believe the the quality of the study was improved.

Q1) Please shorten the title of the article.

A1) The title was shorten requested.

Q2) Please clearly state the purpose of this research.

A2) Purposes of the study were partially re-written

Q3) Please give a hypothesis

A3) Hypothesis was added at the bottom of introduction section.

Q4)Please provide the bioethics committee number. The study is a retrospective medical note revision, no drugs or any experimentation was performed. The cohort differentiation was performed according to the patient characteristics, not for different treatments. In our institution, the bioethics Committee does not request any approval and does not release an approval number for this kind of study.

Q5) The research results should be expanded.

A5) The results discussion was amplified in discussion section

Reviewer 2 Report

I have no relevant comment. I would change tables rows and columns (Now it is illegible).

Very good job, congratulation.

Author Response

Q1) I have no relevant comment. I would change tables rows and columns (Now it is illegible).

Very good job, congratulation.

A1) Thanks for your revision. The tables were re-designed, and now are easier to read.

Reviewer 3 Report

Congratulations for the manuscript. Distal radius fracture is a common pathology also in children, that's why any new analyses should be interesting. This study must be improved in some details:

1.- DEFINE AND CHECK THE THIRD OBJETIVE, it sound to be a mix of the two previous ones.

2.- Make easy results with some figures.

3.- Discussion:

  • It is necessary to analyze the relationship between worse initial translation and worse quality of closed reduction to other risk factors for instability.
  • do the obesity patients have any consequences after final growth-up?
  • does obesity relate with biological bone maturation?
  • and with leptin level?

Author Response

Congratulations for the manuscript. Distal radius fracture is a common pathology also in children, that's why any new analyses should be interesting. This study must be improved in some details:

Q1) DEFINE AND CHECK THE THIRD OBJETIVE, it sound to be a mix of the two previous ones.

A1) Purposes of the study were partially re-written

Q2) Make easy results with some figures.

A2) In order to promote the proper understanding of the paper the table were re-designed and figure added in the manuscript

3.- Discussion:

Q3) It is necessary to analyze the relationship between worse initial translation and worse quality of closed reduction to other risk factors for instability.

A3) Thanks for your comment. We agree with you, in another article on the same topic we investigate the relationship between the worse initial translation and worse quality of closed reduction. This manuscript is focused on obesity as a risk factor, we believe that the analysis of other variants could be misleading. We added a period in discussion

Q4) do the obesity patients have any consequences after final growth-up? does obesity relate to biological bone maturation? and with leptin level?

A4) Thanks for your comment.  We agree with you, but the study design is limited to a retrospective radiological assessment and limited to a radiological schedule. Moreover, simple X-rays do not permit the evaluation of bone maturation and leptin level. A period was added in the limits of the study.

Round 2

Reviewer 1 Report

I accept the amended article

Reviewer 3 Report

Congratulations

The manuscript has been improved successfully